# VISUAL HAYSTACKS: A VISION-CENTRIC NEEDLE-IN-A-HAYSTACK BENCHMARK

**Tsung-Han Wu, Giscard Biamby, Jerome Quenum, Ritwik Gupta,**
**Joseph E. Gonzalez, Trevor Darrell, David M. Chan**
University of California, Berkeley

## ABSTRACT

Large Multimodal Models (LMMs) have made significant strides in visual question-answering for single images. Recent advancements like long-context LMMs have allowed them to ingest larger, or even multiple, images. However, the ability to process a large number of visual tokens does not guarantee effective *retrieval* and *reasoning* for multi-image question answering (MIQA), especially in real-world applications like photo album searches or satellite imagery analysis. In this work, we first assess the limitations of current benchmarks for long-context LMMs. We address these limitations by introducing a new vision-centric, long-context benchmark, "Visual Haystacks (VHs)". We comprehensively evaluate both open-source and proprietary models on VHs, and demonstrate that these models struggle when reasoning across potentially unrelated images, perform poorly on cross-image reasoning, as well as exhibit biases based on the placement of key information within the context window. Towards a solution, we introduce MIRAGE (Multi-Image Retrieval Augmented Generation), an open-source, lightweight visual-RAG framework that processes up to 10k images on a single 40G A100 GPU—far surpassing the 1k-image limit of contemporary models. MIRAGE demonstrates up to 13% performance improvement over existing open-source LMMs on VHs, sets a new state-of-the-art on the RetVQA multi-image QA benchmark, and achieves competitive performance on single-image QA with state-of-the-art LMMs. Our dataset, model, and code are available at: https://visual-haystacks.github.io.

## 1 INTRODUCTION

Large Multimodal Models (LMMs) have demonstrated remarkable success in visual question-answering tasks on single images. Recent advancements, such as long-context LMMs, now allow these models to ingest multiple images simultaneously (Liu et al., 2024a; Chen et al., 2024; Ye et al., 2024). However, fitting more visual tokens within a context window does not inherently translate to improved performance in more complex, multi-image question-answering (MIQA) scenarios. In fact, we demonstrate in Sections 2.1 and 3 that long-context LMMs are unable to accurately *retrieve* or *reason* across thousands of images, thereby debilitating for real-world applications such as searching through photo albums or analyzing medical and satellite imagery. This motivates the development of rigorous benchmarks that evaluate these capabilities, and methods that can appropriately retrieve and reason, in large-scale settings.

The Needle-In-A-Haystack (NIAH) benchmark (Kamradt, 2023) is a popular method for evaluating long-context models. The task is simple: a specific set of words (a "needle") is inserted into a large set of text (a "haystack"). Models are then assessed on their ability to find the needle in the haystack. In natural language processing (NLP), this simple evaluation has revealed intriguing behaviors, such as the "lost-in-the-middle" phenomenon in which needles placed in the middle of the haystack are harder to find than those placed at the beginning or end (Liu et al., 2024b). The utility of this diagnostic has proven difficult to replicate in the vision domain; existing visual NIAH benchmarks (Reid et al., 2024) showcase near-perfect performance when evaluated on a variety of models, leading to limited insights into the models' capabilities. This performance saturation is likely due to the fact that these benchmarks simulate unrealistic scenarios, such as when the "visual" needles are simply out-of-distribution domains with overlaid text serving as the needle (Figure 1 (A)), testing a model's optical character recognition (OCR) capabilities rather than useful object retrieval or reasoning capabilities.

We address these shortcomings by introducing "Visual Haystacks (VHs)," a realistic, simple-to-understand, vision-centric NIAH benchmark that challenges models to find truly "visual needles" and reason about them

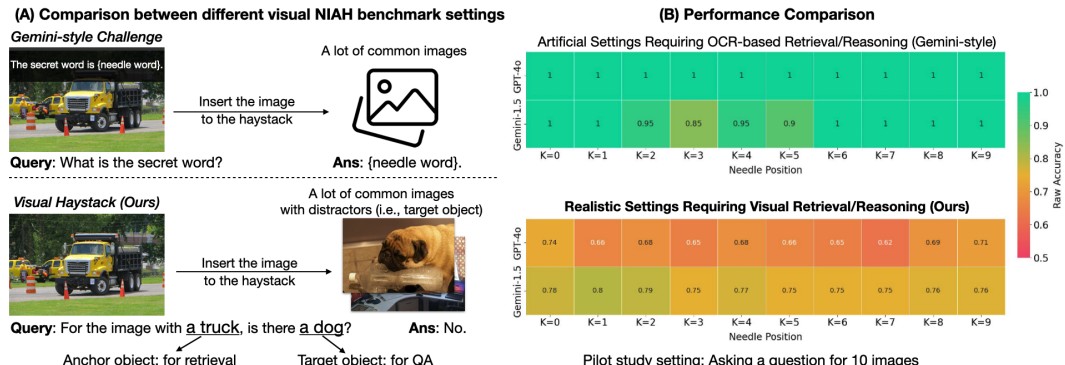

Figure 1: (A) Unlike existing visual Needle-In-A-Haystack (NIAH) challenges (Reid et al., 2024) that overlay needle information as text onto an image, our "Visual Haystacks" (VHs) benchmark is vision-centric, requiring the model to first retrieve the needle image(s) from the haystack and then reason about the image(s) to answer the question. (B) We benchmark existing LMMs under different NIAH settings where only one needle image is present among ten images. While traditional visual NIAH challenges overemphasize text retrieval, which can be easily hacked by state-of-the-art models with strong OCR capabilities, they are unable to solve the simple visual questions in VHs.

among natural, visually distinctive objects. Unlike previous benchmarks, LMMs must integrate and reason about visual information spread across multiple images to perform well on VHs. For example, in Figure 1 (A), a query such as "For the image containing a truck, is there a dog?" requires LMMs not only to *retrieve* the relevant image containing a truck but *reason* about the presence of a dog in this image to answer correctly.

Our evaluation of both open-source and proprietary LMMs on VHs yielded three notable findings: (1) While most LMMs can process up to 100 images, they exhibit up to a 40% performance drop on VHs compared to non-retrieval QA tasks, implying that LMMs struggle with retrieval and are prone to visual distractors. (2) This performance decline is even more severe when multiple images are used as needles, suggesting that LMMs struggle to reason across multiple images. (3) We observed significant positional bias in LMMs, reminiscent of the "lost-in-the-middle" pathology found in NLP tasks (Liu et al., 2024b). Models' sensitivity to the placement of key information within the context window often results in sub-optimal performance when the needle is positioned unfavorably.

In support of developing models which can solve long-context MIQA tasks, we further introduce MIRAGE (Multi-Image Retrieval Augmented Generation), a visual-RAG framework that gives LMMs the capacity to solve large-scale MIQA tasks such as VHs. Built on the LLaVA architecture (Liu et al., 2023a), MIRAGE builds on the strengths of several methods to mitigate context length limitations found in contemporary LMMs, ultimately enabling MIQA on over 10k images. First, we introduce a compressive image encoding strategy to make efficient use of a fixed context window. Second, we implement a query-aware, retrieval-based relevance filter focuses the model on pertinent content, facilitating the processing of larger image sets. Finally, we constructed a 1.2M-image instruction-tuning dataset for MIQA tasks using a mixture of synthetic and real-world data. Evaluations on VHs and other MIQA benchmarks, such as RetVQA (Penamakuri et al., 2023), demonstrate that MIRAGE outperforms existing retrieval-augmented methods and LMM-based approaches in most tasks. Additionally, MIRAGE shows competitive performance on conventional single-image QA tasks compared to other state-of-the-art models. In sum, our contributions are as follows:

- We introduce a new benchmark, "Visual Haystacks (VHs)" which explicitly tests MIQA models on their ability to *retrieve* and *integrate* visual information.
- We systematically assess existing open and closed-source LMMs on the VHs benchmark, and reveal three key findings: susceptibility to visual distractors, difficulty in multi-image reasoning, and a bias in image positioning.
- We introduce a novel baseline for VHs, MIRAGE (Multi-Image Retrieval Augmented Generation), which is the first open-source visual-RAG framework capable of scaling to over 10k images.

## 2 THE VISUAL HAYSTACKS BENCHMARK (VHS)

The Needle-In-A-Haystack (NIAH) evaluation Kamradt (2023) has recently become a widely adopted unit test for evaluating LLM/LMM systems' ability to uniformly process long-context inputs. In the vision

domain, however, no visual NIAH benchmark (Reid et al., 2024) tests whether a model can locate key visual information within a large pool of images and subsequently use that information to answer a specific query.

The predominant "visual" NIAH benchmarks have several limitations as illustrated in Figure 1 (A). For example, Gemini-v1.5's demo creates "visual" needles by overlaying text reading "The secret is `Needle`" on a specific frame in a long video, then asking the model to retrieve this text. While this setup technically evaluates a model's ability to handle long-context inputs, it does so by placing disproportionate emphasis on textual/OCR tasks, thereby underplaying the importance of *image-based retrieval and reasoning*. Additionally, the benchmark includes only a single test case—with a single needle in the video—limiting its ability to comprehensively evaluate models across varied and complex visual scenarios reflective of real-world applications.

We address these shortcomings in the Visual Haystacks (VHs) benchmark which emphasizes realistic, visually grounded needles and mirrors real-world visual long-context learning tasks. The design of VHs is based on two core principles: realism and reliability. The needles in VHs are real objects within natural images, overcoming the artificial inclusion of overlaid text or image patches (Reid et al., 2024; Wang et al., 2024c) that previous visual NIAH benchmarks suffer from. VHs utilizes in-domain images and straightforward questions with human-annotated ground truth. Compared to recent LLM and LMM datasets (Gema et al., 2024; Zhu et al., 2024) which face noises, biases, and out-of-distribution (OOD) challenges due to their data labeling methodology, VHs is much more reliable. Finally, VH is diverse and large-scale; haystack sizes are as large as 10,000 images, haystacks can feature one or more needles, and each needle can draw from over 50 distinct objects. With over 1,000 examples of each setting, VHs provides 97k images total for comprehensive visual NIAH evaluation.

## 2.1 BENCHMARK CONSTRUCTION

We construct the VHs dataset from the COCO dataset (Lin et al., 2014) which has accurate, object-level annotations. To generate a question/answer pair, we first select two objects from COCO's label set to serve as an anchor and target, respectively. These objects then seed question generation in two settings: a single-needle setting with the template "`For the image with anchor object, is there target object?`" and a multi-needle setting with either "`For all images with anchor object, do all of them contain target object?`" (requiring the model to retrieve and look at all relevant anchor images) or "`For all images with anchor object, do any of them contain target object?`" as templates. The answers are binary (yes/no). We curate the dataset such that guessing or relying on common sense reasoning without viewing the image results in a 50% accuracy rate. This design ensures that the anchor object serves as the key for image retrieval, while the target object forms the basis of the actual question during image reasoning.

After selecting the object pairs, setting the questions, and determining the answers, we then compile corresponding image haystacks by first curating the needle images, which contain the "anchor object" as indicated by the paired answer. We then accompany these images with multiple negative distractors to form haystacks of varying sizes, ranging from 1 to 10,000 haystack images. We select distractor images in a manner such that no distrator contains any anchor object (following COCO's object annotations). However, *some of them contain the target objects* so as to create meaningful distractors. The single-needle setting contains only one needle image in the haystack, while the multi-needle setting includes either two and three needle images. VHs consists of 1000 question-answer pairs for both single- and multi-needle settings, with an explicit small subset VHs_small consisting of 100 questions, which is helpful for economical evaluation of expensive closed-source models. Appendix B illustrates several examples and statistics of single- and multi-needle benchmarks.

As shown in Figure 1, the VHs dataset aligns more closely with real-world scenarios compared to previous visual NIAH challenges and challenges contemporary LMMs (section 3). While developing a comprehensive MIQA benchmark with more diverse images and questions beyond COCO is a valuable direction for future research, given the consistently low performance of current LMMs on the VHs benchmark, we believe that it is eminently important to establish this simple foundational unit test for evaluating long-context visual retrieval and reasoning.

## 3 CAN LMMs HANDLE LONG-CONTEXT VISUAL INPUTS?

We evaluated a range of state-of-the-art open-source and proprietary LMMs on the VHs benchmark. Specifically, this includes GPT-4o (OpenAI, 2024), Gemini 1.5 Pro (Reid et al., 2024), Qwen2-VL (Wang

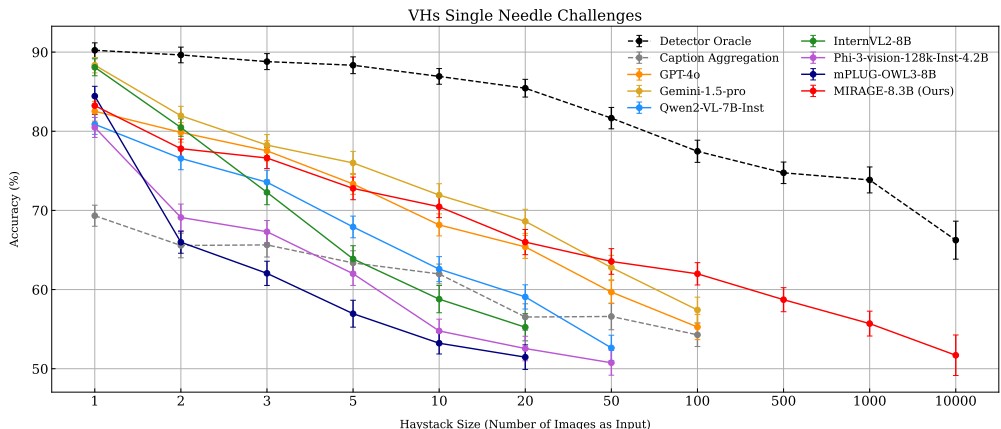

Figure 2: Experimental results on the VHs single-needle challenge. All LMMs experience significant falloff as the size of the haystack (N) increases, indicating that existing approaches are not robust to complex visual-linguistic processing over long visual contexts. Note the non-linear x-axis in this plot.

et al., 2024b), Phi-3 Vision (Abdin et al., 2024), mPLUG-Owl3 (Ye et al., 2024), and InternVL2 (Chen et al., 2024).

We implement two non-LMM baseline approaches to further contextualize LMM performance. The first baseline, termed "caption aggregation," follows a two-stage process in which images are first captioned using Qwen2-VL, and subsequently, Llama-v3.1 (Dubey et al., 2024) is used to answer questions based on the aggregated captions. This is a query-unaware baseline given that this method generates captions without conditioning on the questions being asked, capturing a sub-optimal, yet informative, comparison point. The second baseline, the "detector oracle," serves as an upper-bound baseline by utilizing OWLv2 (Minderer et al., 2024) as an object detection model to first find all images with the anchor object which are then each checked for the target object. Detailed descriptions of both baselines are available in subsection C.1.

In both single-needle and multi-needle settings, we conducted experiments using the full VHs dataset where the haystack size was 100 images or fewer, and switched to the VHs_small subset with larger haystacks to mitigate computational costs. We report the results as bootstrapped averages, with standard deviations indicated as error bars to ensure robustness. The outcomes of the single- and multi-needle challenges are presented in Figure 2 and Figure 3. Figure 4 explores LMM behavior when the needle image is located in different positions within the input image set. Comprehensive results are shown in subsection C.2.

Our analysis yielded three key insights into LMM behavior on long-context visual retrieval and reasoning tasks: (1) a susceptibility to visual distractors, (2) difficulty in reasoning across multiple images, and (3) a bias in relative image positioning. These findings are not possible to conclude using prior visual NIAH benchmarks that utilize artificial needle generation, further highlighting the significance and contribution of our realistic, vision-centric benchmark. We elaborate on each of these observations below.

## 3.1 SUSCEPTIBILITY TO VISUAL DISTRACTORS

As shown in Figure 2, the single-needle challenge reveals that LMMs are significantly impacted by visual distractors. When presented with only one image, general-purpose LMMs such as InternVL2-8B and Gemini v1.5 Pro perform comparably to specialized detectors like OWLv2, indicating that these models can handle simple visual reasoning tasks in isolation. However, as the number of input images increases, a notable drop in performance is observed compared to an oracle baseline that combines a detector with a language parser. This degradation, which does not appear in prior artificial OCR-based NIAH benchmarks as shown in Figure 1 (B), suggests that the main limitation of current LMMs lies in their ability to retrieve relevant visual information from large sets of images containing distractors. This challenge is particularly pronounced in open-source models like InternVL2-8B, which excel at single-image tasks but struggle as the visual context grows in complexity.

An intriguing outcome arises from the caption aggregation baseline: despite being an inherently sub-optimal approach due to the lack of context-specificity when generating captions, it begins to match or even surpass the performance of open-source LMMs when the number of images reaches 20. This indicates that while language models can exhibit robustness against irrelevant text (captions in this case), general LMMs nowa-

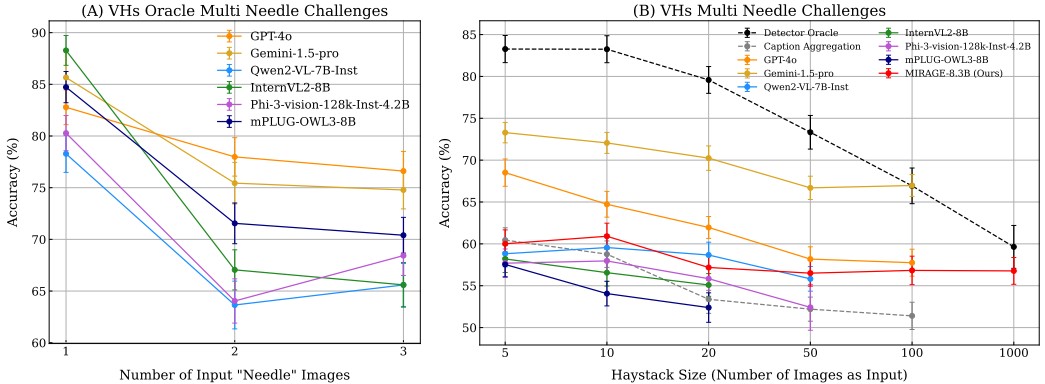

Figure 3: Experimental results on the VHs multi-needle challenge reveal insightful outcomes. (A) The oracle experiment, which uses only needle images as input, demonstrates significant performance degradation in both proprietary and open-source LMMs when required to integrate information across multiple images. (B) In the full multi-needle challenge that includes distractor images, we observed a performance decline of existing LMMs as the size of the haystack (N) increases. Given the same haystack size, the performance deteriorates considerably compared to the single-needle challenge across all models in most scenarios. These findings indicate that current methodologies struggle with real-world, large-scale multi-image QA tasks that demand both visual retrieval and reasoning across extensive visual contexts.

days are still impacted by irrelevant images. Nevertheless, the caption aggregation method is impractical for general multi-image question answering due to its high computational overhead and limited context length. For instance, processing 100 images for a single question can take over three minutes (see Figure 6 for runtime analysis), and 1,000 detailed captions would exceed Llama-v3.1's 128k context length limit.

Beyond the issue of distractors, context length limitations remain a fundamental bottleneck for all tested LMMs. Qwen2-VL, e.g., is constrained by its shorter context windows and can ingest no more than 50 images. While Phi-3 theoretically offers a higher context capacity, it exhausted the memory of four 40GB A100 GPUs when processing 100 images. API-based models such as Gemini v1.5 Pro and GPT-4o can process up to 100 images but cannot solve larger haystacks (1,000+ images) due to API limitations.

## 3.2 DIFFICULTY IN REASONING ACROSS MULTIPLE IMAGES

To further examine LMMs' ability to integrate information across multiple images, we conducted experiments using the VHs multi-needle challenge. This task requires LMMs to first retrieve relevant visual information and then integrate it across images to generate a correct response.

Our initial experiment focused on an oracle cross-image reasoning scenario. In this setting, models were provided only with needle images and a standardized multi-image question template described in subsection 2.1. Interestingly, we observed a sharp performance decline across all LMMs as the number of needle images increased from one to two (Figure 3 (A)). However, as the number of needle images increased from two to three, the performance decline became much smoother or even stabilized. This trend was especially pronounced in open-source LMMs, which exhibited larger performance drops initially. We hypothesize that this may be due to the models being primarily trained on single-image question-answering datasets, with limited exposure to multi-image tasks—an observation corroborated by their technical reports. Overall, these experimental results suggest that LMMs continue to struggle with integrating information across multiple images, even in scenarios where the task of visual retrieval is mostly excluded.

In the full VHs multi-needle challenge, where models must both retrieve visual information and reason across multiple images, the results (shown in Figure 3 (B)) were consistent with those from the single-needle experiments. As the number of input images increased, performance steadily declined across all models. While Gemini v1.5 Pro maintained relatively high accuracy (above 65%, with only a 5% decline from N=5 to N=100), other models experienced steeper drops, with some falling below the caption aggregation baseline. It is particularly noteworthy that GPT-4o, which performed best in the oracle multi-needle experiments with two or three needles, struggled significantly in real-world multi-needle challenges with distractor images, performing worse than Gemini v1.5 Pro by a large margin. Open-source models also demonstrated performance declines from the oracle scenario to the real-world multi-needle challenges, with mPLUG-Owl3 dropping from the top-performing open-source model to the worst among

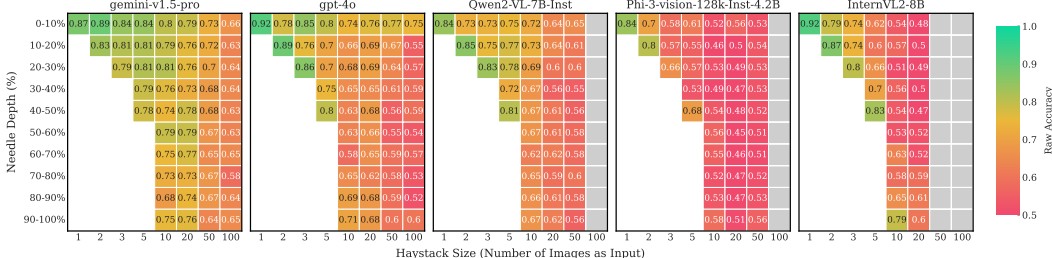

Figure 4: Plots showing needle position, vs. performance on the VHs benchmark for several image settings. For existing LMMs, the needle position is extremely important, with performance degradation of up to 25% when the needle is not placed in the optimal location in the input context. The gray boxes indicate that these experiments exceed the available context length or the model is unable to execute on 4 A100 GPUs.

its peers. Overall, these results highlight the persistent challenges that LMMs face in managing complex, multi-image visual retrieval and reasoning tasks. More in-depth analyses are reported in subsection C.4.

## 3.3 POSITIONAL BIASES

In natural language tasks, prior studies have identified the lost-in-the-middle phenomenon in models like GPT-3.5, where performance deteriorates on long-context retrieval and reasoning tasks when key information is positioned near the middle of the context window. However, previous benchmarks in LMMs did not observe this effect, likely because the datasets were too simple, allowing LMMs to easily retrieve relevant information regardless of its position. In VHs, we revisit this issue by systematically evaluating both open-source and proprietary models using the VHs$_{small}$ single-needle dataset.

As shown in Figure 4, LMMs are also sensitive to the positional placement of the needle image. Interestingly, this positional bias varies across models. For example, Gemini v1.5 Pro tends to favor images at the beginning of the sequence, while GPT-4o exhibits a clear lost-in-the-middle effect, similar to what has been observed in natural language tasks (Liu et al., 2024b). In contrast, open-source models like Qwen2-VL, Phi-3-vision, and InternVL2 favor images closest to the question (i.e., the last image in the sequence) given a small number of images ($N \leq 5$). We break down the 2D heatmap for each method in Appendix C.

## 4 MIRAGE: MULTI-IMAGE RETRIEVAL AUGMENTED GENERATION

In the previous section, we discussed several sub-optimal behaviors of current Large Multimodal Models (LMMs) when handling long-context visual retrieval and reasoning tasks. A particularly pressing challenge is the inability of existing methods to efficiently process large-scale visual inputs—those involving over 1,000 or even 10,000 images. To address this limitation, we introduce **MIRAGE**, an open-source Retrieval-Augmented Generation (RAG) framework for large-scale, long-context visual retrieval and reasoning.

The design of MIRAGE was informed by three guiding principles: (1) Beyond simply extending context length, we aimed to optimize efficiency by reducing the number of tokens per image, enabling the processing of larger sets of images. (2) To minimize computational overhead and reduce vulnerability to irrelevant images, we integrated a lightweight retriever module trained alongside the LMM to filter out distractors—a core RAG approach. (3) Recognizing the scarcity of multi-image QA datasets with realistic visual distractors, we also construct a custom instruction-tuning dataset to enhance the model's retrieval and reasoning capabilities. The following subsections detail these components.

Despite its simplicity, MIRAGE is the first general framework to enable visual-RAG operations with LMMs. The collaboration between a retriever and a processor plays a crucial role in real-world, large-scale multi-image QA scenarios, such as photo album searches or geospatial image analysis. By filtering images based on relevance before reasoning occurs in the LMM, the retriever ensures efficient runtime (see Figure 6) and prevents the LMM from being affected by visual distractors—a known limitation of current LMMs. Furthermore, the LMM, trained with our custom dataset, ensures competitive performance across diverse and general question-answering tasks, as demonstrated in section 5.

### 4.1 MODEL ARCHITECTURE

MIRAGE is a RAG-based solution built on LLaVA-v1.5-7B (Liu et al., 2023a) and Llama-v3.1-8B (Dubey et al., 2024). The model input consists of a series of image tokens followed by a textual question, formatted

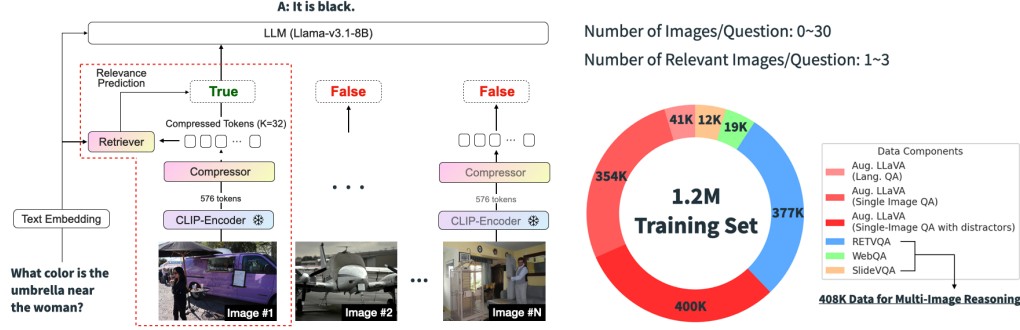

Figure 5: MIRAGE enables large-scale long-context visual retrieval and reasoning through a combination of key components and a large-scale training set. (A) MIRAGE processes questions and images in several stages: first, image features are encoded using CLIP, followed by compression through our Q-Former. A retriever module then calculates relevance scores, ensuring that only the most relevant images are passed to the LLM for final reasoning. The red dashed line illustrates the key difference between conventional LMMs, like LLaVA, and our visual-RAG approach. (B) To further improve MIRAGE's performance in long-context retrieval and reasoning, we introduce a large-scale MIQA instruction-tuning dataset, blending both synthetic and real-world data.

as "`<img_token> <img_token> ... \n <Actual Question>`". Each image is first passed through a frozen CLIP image encoder to extract patch-level features (576 tokens). To reduce the computational and context-length burden, we apply an image compressor to reduce the number of tokens per image. Specifically, we employ a Q-Former (Li et al., 2023b), a lightweight transformer with $K=32$ learned query vectors that cross-attend to the full 576 patch features, reducing the tokens per image from 576 to 32—a significant 18x reduction in token intensity. Following this, the output is passed through two MLP layers with GELU activation functions (Hendrycks & Gimpel, 2016) to align with the LLM's input dimensions. MIRAGE is visualized in Figure 5. Additional comparisons of this compression technique against alternative methods can be found in Table C.2.

Despite token compression allowing MIRAGE to handle over 1,000 images, processing 10,000 images remains inefficient if a significant portion of those tokens are irrelevant to the query. To address this, we implemented a retriever module that performs a lightweight relevance-based filtering of images, ensuring that only pertinent images are processed by the LMM. Although many forms of retrieval can be used (e.g., CLIP similarity thresholds), we found it more effective to use a query-aware retrieval model trained alongside the next-token prediction task (shown in Figure 6, and discussed in section 5). Formally, given a set of images, $I$ made up of image tokens $\{i_0,...,i_n\}$, it is our goal to retain the minimal subset $I_{min} \subset I$ such that we can still accurately answer the query $Q$. Let the image encoding module as a function $\psi(I)$, we can get:

$$F_i, R_i = \psi(I_i, Q), \tag{1}$$

where $F_i$ represents image features and $R_i$ represents the relevance score of the image of Image $I$. In practice, our retriever module consists of several transformer blocks on the query and compressed image features, followed by a sigmoid activation to predict a 0/1 relevance score.

After token reduction and retrieval filtering, the LLM processes the resulting set of visual features along with the encoded question. Like other LMMs, MIRAGE concatenates aligned image features and text features together and passes to the downstream LLM, going through next-token prediction to generate textual output.

## 4.2 MODEL TRAINING

**Training Data:** Due to the limited availability of multi-image QA (MIQA) datasets, we created a training dataset for MIRAGE by combining two main sources: (1) existing MIQA datasets, and (2) synthetic MIQA data derived from single-image QA datasets. We first included all publicly available MIQA training sets, including RetVQA (Penamakuri et al., 2023), SlideVQA (Tanaka et al., 2023), and WebQA (Chang et al., 2022). RetVQA, derived from Visual Genome, contains 377K questions but focuses on narrow domains such as object attributes, relationships, and counting. SlideVQA and WebQA add some diversity but are limited in size, containing only 19K and 12K questions, respectively. These datasets typically include one or two relevant images out of a set of 15-30 images.

To supplement the training data, we adapted LLaVA's single-image QA data into a multi-image format. Instead of simply adding random distractors, which risks diluting the relevance of questions, we employed

| Method | RetVQA | VQAv2 | GQA | TextVQA | POPE | MMB | MMB-CN | MME | SEED | MM-Vet |
|---|---|---|---|---|---|---|---|---|---|---|
| GPT-4o | 34.6 | 77.2 | - | 78.0 | 87.2 | - | - | 1614.2 | - | - |
| Gemini v1.5 Pro | 32.2 | 73.2 | - | 73.5 | 88.2 | - | - | 1562.4 | - | - |
| LLaVA-v1.5-7B | 30.6 | 78.5 | 62.0 | 58.2 | 85.9 | 64.3 | 58.3 | 1510.7 | 58.6 | 31.1 |
| LWM | - | 55.8 | 44.8 | 18.8 | 75.2 | - | - | - | - | 9.6 |
| MIRAGE-8.3B (Ours) | 67.6 | 76.6 | 59.1 | 56.2 | 85.4 | 69.2 | 66.9 | 1437.9 | 59.0 | 33.4 |

Table 1: Comparative performance of methods on multi-image and single-image QA tasks. Performance metrics for single-image QA baselines are sourced from their official papers or the online leaderboard. MIRAGE shows strong performance in multi-image QA and competitive performance in single-image QA compared to existing proprietary and open-source long-context models. Built primarily on LLaVA-v1.5-7B, MIRAGE-8.3B matches its performance on single-image QA while excelling in multi-image QA, achieving this with an 18x reduction in tokens per image.

a keyword-based filtering method to cluster questions with similar content, followed by random sampling of two to ten distractor images from unrelated subsets. This process results in an instruction-tuning dataset containing around 600K samples. We further shuffle the images within each instruction-tuning pair during training to ensure the relevant images can appear in any position, forcing the model to remain insensitive to the positional context of relevant images. Figure 5 (B) shows the composition of the assembled dataset.

**Training Procedure:** MIRAGE's training follows a two-stage process: pre-training and fine-tuning. During pre-training, the CLIP visual encoder and LLM backbone are frozen, with only the Q-Former and MLP layers trained on the next-token prediction task. For pre-training, we utilized a combination of ShareGPT captioning data (Chen et al., 2023), alongside large-scale datasets like CC3M/12M (Changpinyo et al., 2021), LAION-400M (Schuhmann et al., 2022), and COCO (Lin et al., 2014), using a data annealing approach (Li et al., 2024) to emphasize high-quality data in the later stage of the training. During fine-tuning, the retriever module is activated, and the entire model (except the CLIP encoder) is trained on both real and synthetic MIQA data. In addition to the next-token prediction task, we co-trained the retriever using the binary cross-entropy loss, assigning a higher weight (5.0) to positive samples to address data imbalance and prioritize recall. The instruction tuning was completed in two days using 16 A100 GPUs, with the first 60% of the training focused on passing only relevant images to the LLM. In the remaining 40%, several distractor images were added to improve robustness, following recommendations from (Zhang et al., 2024).

## 5  RESULTS & DISCUSSION

**Visual Haystacks:** MIRAGE's performance on single- and multi-needle VHs challenge is presented in Figure 2 and Figure 3. All baselines and MIRAGE are tested in an identical, zero-shot setting in which they are not trained on these kinds of questions. While MIRAGE operates as a visual-RAG framework with a retriever and an LMM, it is fundamentally an end-to-end trained "single-model architecture" (Figure 5). In this context, the "retriever" component should be understood as a component for filtering irrelevant images and enhancing long-context reasoning. Based on this, we believe that our comparison with other "single-model LMMs" on VHs is both fair and valid. In the single-needle challenge, MIRAGE demonstrates competitive oracle (N=1) performance against existing approaches. Notably, MIRAGE is the only method that scales to 10k input images, outperforming all open-source models when processing more than three images, and surpassing both Gemini v1.5 Pro and GPT-4o when using over 50 images.

In the multi-needle setting, in addition to handling a significantly larger number of input images, MIRAGE outperforms open-source LMMs in most cases and approaches GPT-4o's performance when using 100 images. However, like other open-source models, MIRAGE exhibits a drop in performance when transitioning from single-needle to multi-needle tasks, especially compared to Gemini v1.5 Pro. These results highlight the need for further advancements in the cross-image reasoning capabilities of these models. We hypothesize that the degraded performance stems from current training datasets which typically focus on two-image inputs and lack diverse question types. This limits MIRAGE's ability to effectively handle tasks requiring reasoning across more than three images with varied queries. Future enhancements to MIQA datasets could address these challenges.

**Conventional VQA Tasks:** Although the primary focus of this paper is on large-scale multi-image QA tasks like VHs, we also evaluate MIRAGE on conventional VQA tasks, including the RetVQA multi-image QA (Penamakuri et al., 2023) and several single-image QA tasks, such as VQA-v2 (Goyal et al., 2017), GQA (Hudson & Manning, 2019), TextVQA (Singh et al., 2019), POPE (Li et al., 2023c), MMB (Liu et al., 2024c), MMB-CN (Liu et al., 2024c), MME (Fu et al., 2023), SEED-Bench (Li et al., 2023a), and

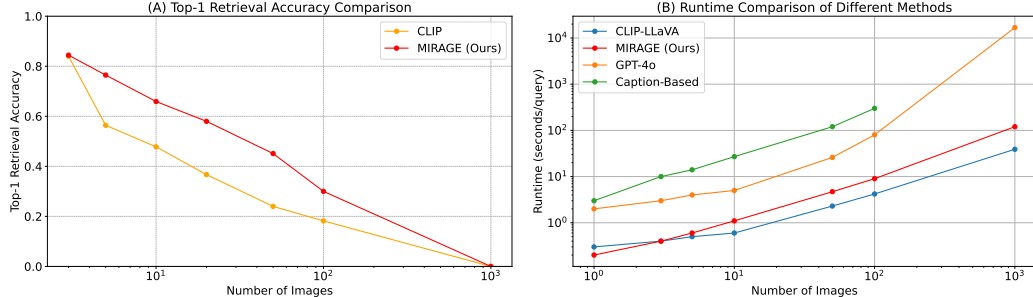

Figure 6: Analyses on MIRAGE and CLIP retriever across performance and runtime. Both approaches with a retrieval component are significantly more efficient than GPT-4o and caption-based baselines. While CLIP is slightly faster, MIRAGE has significantly higher recall, which impacts downstream performance.

MM-Vet (Yu et al., 2023b). Baseline comparisons include GPT-4o (OpenAI et al., 2024), Gemini v1.5 Pro (Reid et al., 2024), LWM (Liu et al., 2024a), and LLaVA-v1.5-7B (Liu et al., 2023a). In all cases, we follow standard evaluation procedures from either RetVQA (Penamakuri et al., 2023) or LLaVA (Liu et al., 2023b). As shown in Table 1, MIRAGE achieves state-of-the-art performance on the RetVQA multi-image QA benchmark. On single-image QA tasks, MIRAGE exhibits minor performance degradation on VQA-v2, GQA, POPE, and TextVQA compared to LLaVA-v1.5-7B, which we attribute to MIRAGE's token compression scheme (See subsection C.3 for more discussions). However, MIRAGE surpasses LLaVA-v1.5-7B on benchmarks like MMB, MMB-CN, and MM-Vet, consistently outperforming LWM across all evaluated tasks. Additionally, MIRAGE remains competitive with proprietary models, making it one of the most versatile open-source frameworks for both single- and multi-image QA tasks.

**MIRAGE retriever vs. CLIP:** While MIRAGE trains the retrieval model inline, it is also possible to use an external retrieval module such as CLIP (Radford et al., 2021) to perform lookup prior to passing images to the downstream LLM. To explore this question, we compare the single-stage MIRAGE retriever with using CLIP as the retriever on the VHs benchmark in Figure 6. This experiment demonstrates that while CLIP is somewhat faster, it struggles with recall, a critical metric for the MIRAGE system, as poor recall leads to necessary information being dropped before even making it to the LLM for analysis. Additionally, we include the performance of CLIP paired with LLaVA-v1.5-7B on single-needle VHs in Table C.3, where it underperforms compared to our MIRAGE-8.3B for almost all cases.

**Conclusion:** The experiments highlight three primary advantages of MIRAGE: *Scalability*, as it is the only framework capable of processing thousands of images in a single query, significantly exceeding the capacity of existing models; *Performance*, with state-of-the-art results on multi-image tasks and competitive results on single-image tasks despite the usage of fewer tokens per image; and *Efficiency*, achieved through token reduction via the retriever component and the co-trained Q-former compressor, allowing MIRAGE to run on a single 40GB A100 GPU with faster runtimes and better retrieval performance compared to CLIP.

# 6 RELATED WORK

Ours is not the first work to address the MIQA problem. Similar to our work, Bansal et al. (2020) introduce a method for multi-image question answering, however in their approach, all of the images (up to 10) are relevant to the question, and their method does not contain a retrieval component. Penamakuri et al. (2023) propose a dataset and method for retrieval-based visual question answering (RetVQA), and share with our work the fact that answers must be gleaned from a set of both relevant and irrelevant images. However, their work differs from ours in two key ways: (1) RetVQA question contexts contain at most two relevant images, meaning that models do not have to reason across many images, largely sidestepping the problem of limited context length, and (2) their approach only allows for small image worlds (up to 30), meaning that they can use a pairwise encoder for each image, and they do not need to search over a large dataset of images potentially containing distractors, limiting the efficiency of their approach. Finally, they are only able to perform question-answering over the images, whereas we pursue a method that can additionally perform more complex reasoning tasks given the images in the dataset.

Similar to our MIRAGE approach are Chen et al. (2022) and Yasunaga et al. (2022) which both retrieve multiple visual documents to answer queries. In the case of Chen et al. (2022), the queries are open-ended question answering, and thus, are not grounded within a particular set of context images. Yasunaga et al. (2022) focuses on one and few-shot classification and image generation, and does not use their multi-image

retrieval to answer aggregated questions about those images. In single-image QA, image retrieval in large multimodal models (LMMs) has been explored using "retrieval-tokens" (Koh et al., 2023), however, it is unclear how such an approach would scale to multi-image QA problems with multiple relevant images. Complementary to our approach, several methods have focused on true long-context LMMs by introducing down-sampling techniques or state-space models (Gupta et al., 2024; Li et al., 2023b; Chen et al., 2024; Abdin et al., 2024; Wang et al., 2024d) - and while these methods cannot yet handle 10K images well, we expect in the future for this to be a promising direction of further research. In addition to general QA, several other works contain domain-specific multi-image QA problems, including Slide VQA (Tanaka et al., 2023), Multimodal QA (Talmor et al., 2021), WebQA (Chang et al., 2022) and Document VQA (Tito et al., 2021).

Outside of VQA, In traditional NLP, retrieving small passages or single documents from large-scale corpora has proven effective. Zhang et al. (2024) introduces a method that fine-tunes LLMs on both relevant and irrelevant documents to support better RAG performance, but does not train an explicit query-aware filter or compression module. ATLAS (Izacard et al., 2023) treats documents as latent variables during training, allowing for efficient retrieval, but requires a complex joint-training setup. Similarly, several methods (Shi et al., 2023; Ram et al., 2023; Borgeaud et al., 2022; Khandelwal et al., 2019) have demonstrated success with zero or few-shot augmentation of standard LLM contexts with retrieved documents. Beyond context augmentation, several traditional NLP approaches (Lin et al., 2023b; Wang et al., 2023; Xu et al., 2023; Liu et al., 2024d; Xiang et al., 2024; Xu et al., 2024) have demonstrated that fine-tuning LLMs to be robust to noisy RAG outputs can lead to performance improvements.

Contemporary to VHs, Wang et al. (2024c) and Zhao et al. (2024) introduce NIAH-like synthetic benchmarks for multi-image and video QA applications, respectively. These datasets focus on short-range retrieval problems (<100 images/frames) and still consist of artificial content like out-of-distribution visual "needles." In contrast, VHs uses realistic, in-distribution visual content in real images and scaling up to 10K images per query to pose a more complex, real-world challenge. Similarly, new benchmarks like CompBench (Kil et al., 2024), MANTIS (Jiang et al., 2024), and MUIRBench (Wang et al., 2024a) were recently proposed to evaluate LMMs' general multi-image reasoning performance. However, these datasets are limited to fewer than 10 images per question and can make diagnosing specific LMM abilities difficult, as answering a single question often requires multiple intermixed capabilities. VHs distinguishes itself by offering a diagnostic, large-scale evaluation explicitly designed to test LMMs' visual retrieval and cross-image reasoning capabilities without such confounding factors.

**MIQA vs. Video-QA:** While several methods have been developed for video question answering (such as Video-LLaVA (Lin et al., 2023a)), the task of answering questions over video is fundamentally different from MIQA. While models that can solve MIQA can often solve video tasks, the inverse is not true, as video frames contain many frame-wise inter-dependencies that are exploited by encoder models (such as MAG-ViT (Yu et al., 2023a) which use temporal blocks, or frame-subsampling, which drops closely related frames). MIQA has independent images, rendering most video models inadequate for this task. It is an interesting direction for future work to explore how to connect MIQA and Video-QA, particularly across the data dimension, wherein Video-QA datasets could provide useful training data for MIQA models.

## 7 CONCLUSION

In this work, we introduce "Visual Haystacks (VHs)", a benchmark specifically designed to challenge LMMs in multi-image question answering (MIQA) by testing their ability to both retrieve relevant images from large collections of unrelated inputs and reason across them. Our evaluation shows that VHs provides significantly more realistic and challenging tasks compared to existing visual NIAH benchmarks which effectively measure OCR or text-based performance. With VHs, we then identified three key limitations across both open-source and proprietary models: susceptibility to visual distractors, difficulty reasoning across multiple images, and biases due to the positional placement of key information within the context window. Additionally, we found that contemporary LMMs are limited to processing no more than 100 images simultaneously. To address these limitations, we introduced MIRAGE, an open-source visual-RAG framework capable of handling up to 10,000 images on a single 40GB A100 GPU. MIRAGE showed superior performance over existing open-source LMMs on VHs and set a new state-of-the-art on the RetVQA benchmark while maintaining reasonable single-image QA performance.

Both VHs and MIRAGE represent some of the first concrete steps towards large models capable of answering questions over thousands or tens of thousands of unrelated images. While they represent strong steps forward, the problem is far from solved, and we hope that this benchmark and framework will inspire continued research in models and the MIQA problem.

ACKNOWLEDGMENTS

We thank Lisa Dunlap, Boyi Li, and Xudong Wang for their invaluable feedback during the discussions. This project was primarily supported by a grant from the National Geospatial-Intelligence Agency (Grant No. HM0476-22-1-2001). Authors, as part of their affiliation with UC Berkeley, were supported in part by the National Science Foundation, US Department of Defense, and/or the Berkeley Artificial Intelligence Research (BAIR) industrial alliance program, as well as gifts from Anyscale, Astronomer, Google, IBM, Intel, Lacework, Microsoft, Mohamed Bin Zayed University of Artificial Intelligence, Samsung SDS, Uber, and VMware. Any opinions, findings, conclusions or recommendations expressed in this material are those of the author(s) and do not necessarily reflect the views of NGA, DoD, or the US government. The paper is approved for public release in accordance with NGA-U-2024-01397.

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

APPENDIX

In this appendix, we include several additional discussions:

- Appendix A provides information on the code release, including links to the codebases and datasets used in the project.

- Appendix B includes additional analyses of the VHs benchmark, discussing the dataset distribution and key characteristics, along with visual examples of single- and multi-needle tasks.

- Appendix C describes the implementation of the baseline models used in the VHs experiments and presents comprehensive results from the VHs experiments, including detailed performance metrics for single- and multi-needle tasks.

- Appendix D explores the potential limitations of the VHs benchmark and the MIRAGE model, along with their societal impacts.

## A   CODE RELEASE

The project page of this paper is at: `https://visual-haystacks.github.io/`. Code for MIRAGE is made publicly available under the MIT license at `https://github.com/visual-haystacks/mirage`, and is derived from the Apache 2.0-licensed LLaVA codebase (Liu et al., 2023b). Code for the VHs benchmark is made publicly available under the MIT license at `https://github.com/visual-haystacks/vhs_benchmark` and contains elements drawn from the MIT-Licensed POPE benchmark (Yifan et al., 2023).

## B   MORE ANALYSES ON VHs BENCHMARK

In subsection 2.1, we previously discussed the construction of the VHs benchmark dataset. Here, we take a closer look at its key aspects to provide a more detailed understanding. Figure B.1 presents the distribution of anchor and target objects within the VHs dataset. Our dataset exhibits a diverse range of objects, and the question/answer pairs are well-balanced. Additionally, we include a Language-Only baseline in Table C.3 to emphasize that the benchmark task cannot be easily solved without leveraging the visual content. These statistics support that our benchmark is both representative and free from bias.

Furthermore, we visualize real single- and multi-needle examples in Figure B.2 and Figure B.3, respectively. These examples highlight the realism and vision-centric nature of our VHs dataset, establishing it as a robust and reliable visual NIAH benchmark.

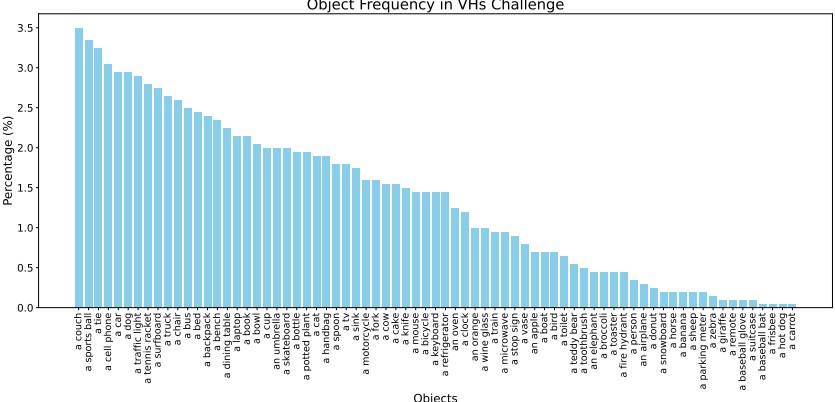

Figure B.1: We visualize the distribution of the anchor (needle) and target objects in the VHs dataset, highlighting its diversity and comprehensive coverage of common objects from COCO.

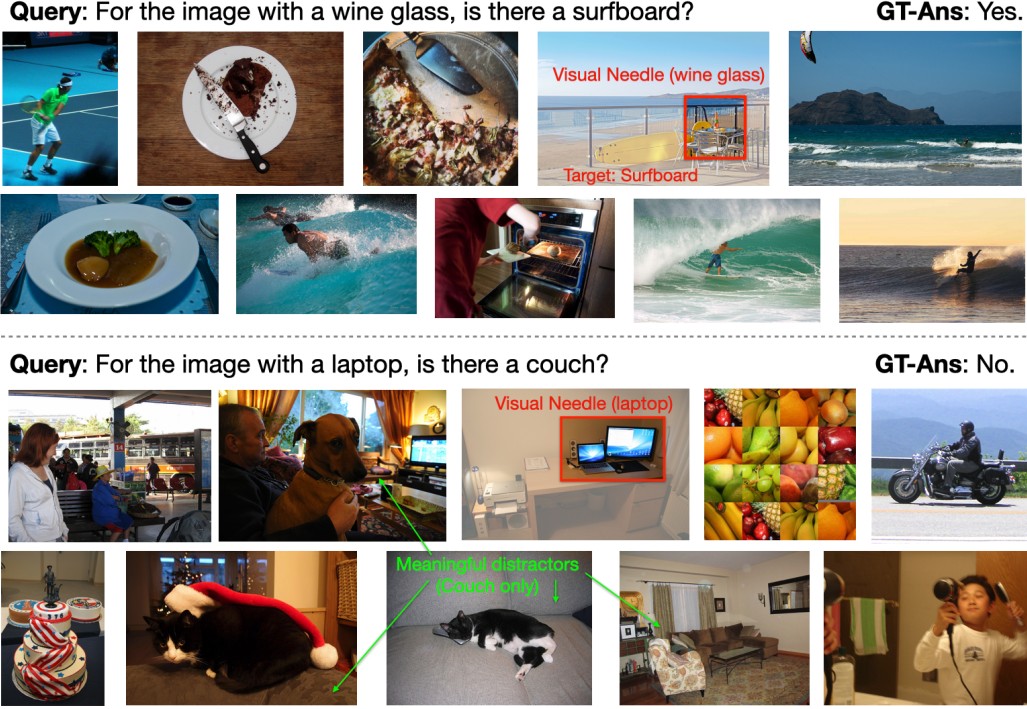

Figure B.2: Real examples of the single-needle VHs challenges (10-image sub-task shown). These visual needles are more realistic and challenging compared to the copy-and-paste approaches in (Reid et al., 2024; Wang et al., 2024c). Models must retrieve the correct image containing the visual needle and answer the associated question. Distractor images are intentionally included to increase the task difficulty.

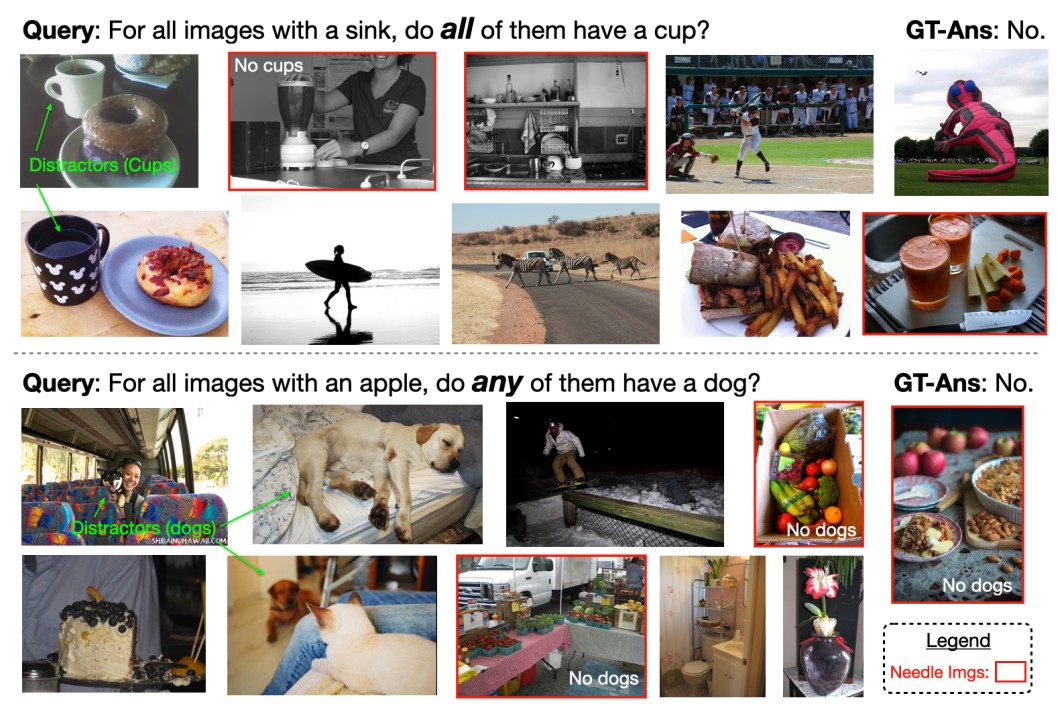

Figure B.3: Real examples of the multi-needle VHs challenges (10-image subtask shown). This task is significantly more difficult than the single-needle task, as it requires models to integrate information across multiple images in addition to handling distractors.

## C    DETAILS ON THE VHS EXPERIMENTS

### C.1    BASELINE IMPLEMENTATION

In section 3, we benchmarked several LMMs and included two non-LMM baselines. Their implementations are detailed below:

**Detector Oracle:** This baseline assumes access to a specialized object detector, specifically the OWLv2 open-vocabulary detector, and an ideal language parser capable of perfectly extracting the anchor object (the visual needle) and the target object. For the single-needle challenge, given a set of images, OWLv2 first detects whether each image contains the anchor object. From the image with the highest detection confidence, we then verify if it also contains the target object and return a yes/no answer. For the multi-needle challenge, we apply a detection confidence threshold and check whether images identified as containing the anchor object also contain the target object. Depending on the question format—whether asking for all or any of the objects—simple control flow determines the final yes/no answer. This setup, leveraging a specialized detector rather than a general LMM and assuming a perfect query understanding, serves as an upper-bound baseline.

**Caption Aggregation:** Another interesting baseline is a non-query-aware approach where we first caption each image and then use an LLM to aggregate these captions to answer the question. In practice, we prompt Qwen2-VL with the instruction, "Please provide detailed and concrete captions of the image," to generate high-quality image descriptions. These captions are then passed to Llama-v3.1-8B, prompted in a specific format (showing in Figure C.1) to integrate the information and provide an answer.

**LMMs:** For proprietary models, we used the APIs for "gpt-4o-2024-05-13" and "gemini-1.5-pro-001." For open-source models, we employed their official Huggingface versions, with details provided in Table C.1. Each model was prompted with: "You are given a set of images. Please answer the following question in Yes or No: {real question}."

```
You are a top expert in interpreting image captions and providing pre-
cise answers to questions based on the information they contain.
Here are the captions for a set of images:

# Caption (1)
{first generated caption}
# Caption (2)
{second generated caption}
...

Based on these image captions, please answer the following question:
{Real Question}.  Please assume there must be at least one image that
satisfies the condition.  Answer with 'Yes' or 'No' only.
```

Figure C.1: The prompt for Llama-v3.1 (8B) used in the caption aggregation baseline.

| Model Name | Hugging Face Model Link |
|---|---|
| OWLv2 (Minderer et al., 2024) | https://huggingface.co/openai/clip-vit-base-patch32 |
| LongVILA-8B-1024Frames (Xue et al., 2024) | https://huggingface.co/Efficient-Large-Model/Llama-3-LongVILA-8B-1024Frames |
| Qwen2-VL-7B-Instruct (Wang et al., 2024b) | https://huggingface.co/Qwen/Qwen2-VL-7B-Instruct |
| Phi3-vision-128k-instruct-4.2B (Abdin et al., 2024) | https://huggingface.co/microsoft/Phi-3-vision-128k-instruct |
| InternVL2-8B (Chen et al., 2024) | https://huggingface.co/OpenGVLab/InternVL2-8B |
| mPLUG-OWL3-8B (Ye et al., 2024) | https://huggingface.co/mPLUG/mPLUG-Owl3-7B-240728 |
| Idefics3-8B-Llama3 (Laurençon et al., 2024) | https://huggingface.co/HuggingFaceM4/Idefics3-8B-Llama3 |
| LLaMA-v3.1 (Dubey et al., 2024) | https://huggingface.co/meta-llama/Llama-3.1-8B-Instruct |

Table C.1: Open-source models and their corresponding HuggingFace links used in the evaluation.

### C.2    COMPREHENSIVE RESULTS

In addition to the line charts in Figure 2 and Figure 3, we provide qualitative results, raw data, and more baseline comparisons in the appendix. Two failure cases of GPT-4o and Gemini v1.5 Pro are shown in Figure C.2. The complete results for the single-needle and multi-needle challenges are presented in Table C.3 and Table C.4, respectively. To better illustrate the positional bias pathology highlighted in Figure 4, we include multiple 1-D plots in Figure C.4 for clearer interpretation. For proprietary models, we observe that Gemini v1.5 Pro tends to favor needle images at the beginning of the sequence, whereas

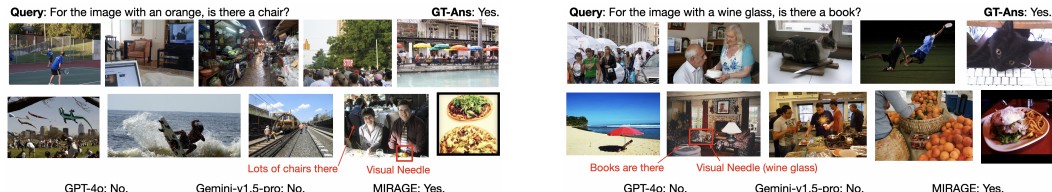

Figure C.2: Qualitative examples on our VHs benchmark between GPT-4o, Gemini-v1.5, and MIRAGE. With a retriever, MIRAGE performs significantly better when dealing with small or non-salient objects.

| Method | Tokens/Img | VQAv2 | GQA | Vizwiz | TextVQA | POPE | MMB | MMB-CN | MM-Vet |
|---|---|---|---|---|---|---|---|---|---|
| Original LLaVA | 576 | 78.5 | 62.0 | 50.0 | 58.2 | 85.9 | 64.3 | 58.3 | 31.1 |
| 3x3 Max-Pooling | 64 | 68.7 | 56.2 | 41.3 | 48.5 | 83.0 | 59.2 | 49.3 | 24.3 |
| Global Avg. Pooling | 1 | 62.5 | 51.3 | 37.7 | 45.5 | 79.6 | 55.0 | 45.5 | 18.9 |
| Q-Former (Ours) | 32 | 72.8 | 56.6 | 48.0 | 47.1 | 83.9 | 61.5 | 55.0 | 27.3 |

Table C.2: Exploration of various token reduction methods. The Q-former demonstrates the most efficiency in reducing token count while maintaining the majority of general QA performance. All experiments were conducted using the official LLaVA-v1.5-7B model with Vicuna-v1.5 as the LLM and LLaVA's instruction tuning data. Note that the Q-former variant shows lower performance compared to our final model MIRAGE-8.3B, which leverages better training data and a more powerful LLM, Llama-v3.1-8B.

GPT-4o shows a preference for images that are not in the middle. In contrast, for open-source models with a small number of images ($N \leq 5$), LMMs perform significantly better when the image is placed near the question (at the end of the image sequence). However, as the number of images increases, models like Qwen2-VL shift their preference towards images at the beginning, while other models still favor images at the end. We believe this intriguing behavior is influenced by the training data, but further investigation is needed to confirm and address this issue in future work.

## C.3 IMAGE TOKEN COMPRESSION SCHEME

As described in subsection 4.1, MIRAGE aims to facilitate future MIQA research with a critical focus on reducing the tokens required per image. To explore this, we evaluated various token compression methods using the LLaVA-v1.5-7B model and their single-image instruction tuning dataset—a simpler setup compared to the full MIRAGE framework with LLama-3.1, a retriever, and our custom data in Table 1. The baseline schemes included max-pooling and average-pooling compression, reducing an image to a single token. While previous work has suggested that Q-Former is sub-optimal (Fan et al., 2024), our results in Table C.2 indicate that Q-Former retains reasonable performance while significantly reducing context length.

## C.4 FURTHER ANALYSES ON MULTI-NEEDLE CHALLENGES

As shown in Figure 2 and Figure 3, we found that the performance on the multi-needle track sometimes surpasses that of the single-needle track for image sets larger than 20. We would like to clarify that the performance on the multi-needle track is not necessarily lower than that of the single-needle track. As mentioned in subsection 2.1, the multi-needle track poses questions in the format: "Q: For all images with <anchor object>, do ALL/ANY of them contain <target object>? A: Yes/No." We can then categorize each data point in the multi-needle track into two cases:

1. The "ANY-Yes" / "ALL-No" QA pairs: These cases are comparatively easier in the multi-needle track since retrieving at least one correct image can suffice. Conversely, the single-needle task demands precise retrieval, posing a greater challenge for models prone to false negatives.

2. The "ANY-No" / "ALL-Yes" QA pairs: These are more difficult in the multi-needle track compared to the single-needle one, as the model must retrieve all relevant images and integrate their information, demanding stronger cross-image reasoning.

The empirical results in Figure C.3 support the above explanation, demonstrating that the benchmark effectively reveals nuanced model behaviors rather than exposing a design flaw. These findings align with those in Figure 3 (A), where LMMs consistently struggle with tasks requiring the integration of information across multiple frames.

| Method | | N=1 | N=2 | N=3 | N=5 | N=10 | N=20 | N=50 | N=100 | N=500 | N=1K | N=10K |
|---|---|---|---|---|---|---|---|---|---|---|---|---|
| Baselines | Language Only (Llama-v3.1) | 49.07%±1.55% | - | - | - | - | - | - | - | - | - | - |
| | Detector Oracle | 90.23%±0.94% | 89.64%±0.99% | 88.80%±1.01% | 88.34%±1.06% | 86.93%±0.99% | 85.44%±1.12% | 81.66%±1.35% | 77.47%±1.40% | 74.75%±1.36% | 73.85%±1.64% | 66.24%±2.40% |
| | Caption Aggregation | 69.33%±1.33% | 65.57%±1.56% | 65.64%±1.53% | 63.37%±1.53% | 61.97%±1.25% | 56.53%±1.67% | 56.60%±1.68% | 54.28%±1.48% | E | E | E |
| Proprietary LMM | Gemini-1.5-pro | 88.35%±0.97% | 81.96%±1.19% | 78.25%±1.33% | 76.00%±1.46% | 71.94%±1.43% | 68.62%±1.52% | 62.78%±1.54% | 57.42%±1.62% | E | E | E |
| | GPT-4o | 82.53%±1.25% | 79.86%±1.23% | 77.53%±1.27% | 73.34%±1.31% | 68.17%±1.39% | 65.39%±1.45% | 59.68%±1.45% | 55.27%±1.60% | E | E | E |
| Open-Source LMM | LongVILA-8B-1024Frames | 63.80%±1.44% | 59.01%±1.71% | 57.73%±1.39% | 56.69%±1.53% | 55.57%±1.61% | 51.99%±1.58% | 52.05%±1.68% | 52.04%±1.51% | E | E | E |
| | Qwen2-VL-7B-Instruct | 80.88%±1.27% | 76.57%±1.43% | 73.56%±1.52% | 67.91%±1.37% | 62.58%±1.58% | 59.07%±1.53% | 52.63%±1.60% | E | E | E | E |
| | Phi-3-vision-128k-inst-4.2B | 80.48%±1.26% | 69.11%±1.69% | 67.31%±1.41% | 62.00%±1.48% | 54.77%±1.49% | 52.55%±1.55% | 50.75%±1.57% | E | E | E | E |
| | InternVL2-8B | 88.08%±1.07% | 80.47%±1.10% | 72.28%±1.56% | 63.87%±1.67% | 58.79%±1.72% | 55.24%±1.72% | E | E | E | E | E |
| | mPLUG-OWL3-8B | 84.44%±1.24% | 65.98%±1.39% | 62.05%±1.53% | 56.95%±1.70% | 53.22%±1.36% | 51.46%±1.55% | E | E | E | E | E |
| | Idefics3-8B-Llama3 | 0.28%±0.18% | 0.37%±0.19% | 0.29%±0.17% | 0.18%±0.13% | E | E | E | E | E | E | E |
| RAG-based | CLIP + LLaVA-v1.5-7B | 85.84%±1.24% | 77.08%±1.43% | 75.75%±1.32% | 68.62%±1.46% | 63.61%±1.62% | 60.35%±1.60% | 55.27%±1.61% | 57.49%±1.45% | 55.43%±1.53% | 52.86%±1.57% | 49.30%±2.20% |
| | MIRAGE-8.3B (Ours) | 83.24%±1.11% | 77.80%±1.21% | 76.62%±1.32% | 72.78%±1.43% | 70.46%±1.37% | 66.00%±1.59% | 63.55%±1.62% | 61.99%±1.41% | 58.72%±1.52% | 55.70%±1.57% | 51.70%±2.56% |

Table C.3: Performance on VHs for single-needle questions. We can observe that as the number of input images (N) increases, LMMs' performance degrades rapidly, specifically for open-source models. This indicates their difficulty in retrieving the correct visual needle from large-scale visual inputs. "E" denotes either exceeding context length or failure to execute on 4 × 80GB A100 GPUs for open-source models, or an API error for proprietary models. Additionally, we observed that the Idefics3 model demonstrated low compliance, often failing to provide a Yes/No response to our questions.

| Method | | N=5 | N=10 | N=20 | N=50 | N=100 | N=1K |
|---|---|---|---|---|---|---|---|
| **Baselines** | Language Only (Llama-v3.1) | 48.79% ± 2.16% | - | - | - | - | - |
| | Detector Oracle | 83.27% ± 1.62% | 83.25% ± 1.61% | 79.58% ± 1.60% | 73.33% ± 2.01% | 66.93% ± 2.13% | 59.64% ± 2.56% |
| | Caption Aggregation | 60.45% ± 1.47% | 58.77% ± 1.58% | 53.38% ± 1.68% | 52.20% ± 1.44% | 51.40% ± 1.63% | |
| **Proprietary LMM** | GPT-4o | 68.51% ± 1.63% | 64.73% ± 1.54% | 61.96% ± 1.31% | 58.18% ± 1.47% | 57.75% ± 1.61% | |
| | Gemini-1.5-pro | 73.29% ± 1.22% | 72.06% ± 1.25% | 70.24% ± 1.46% | 66.69% ± 1.39% | 66.97% ± 1.30% | |
| **Open-Source LMM** | Qwen2-VL-7B-Instruct | 58.82% ± 1.56% | 59.56% ± 1.55% | 58.67% ± 1.53% | 55.82% ± 1.47% | | |
| | InternVL2-8B | 58.20% ± 1.62% | 56.56% ± 1.60% | 55.09% ± 1.36% | | | |
| | Phi-3-vision-128k-Inst-4.2B | 57.69% ± 1.70% | 57.96% ± 1.50% | 55.84% ± 1.39% | 52.42% ± 2.74% | | |
| | mPLUG-OWL3-8B | 57.54% ± 1.47% | 54.06% ± 1.47% | 52.39% ± 1.76% | | | |
| **RAG-based** | MIRAGE-8.3B (Ours) | 60.01% ± 1.66% | 60.91% ± 1.56% | 57.18% ± 1.47% | 56.50% ± 1.55% | 56.83% ± 1.70% | 56.77% ± 1.60% |

Table C.4: As in the multi-needle experiments, the performance of LMMs degrades rapidly as the number of input images (N) increases. The low performance of most LMMs at the beginning (N=5) highlights their difficulty in both retrieving the correct images and integrating information across frames. The definition of "E" is the same as in Table C.3.

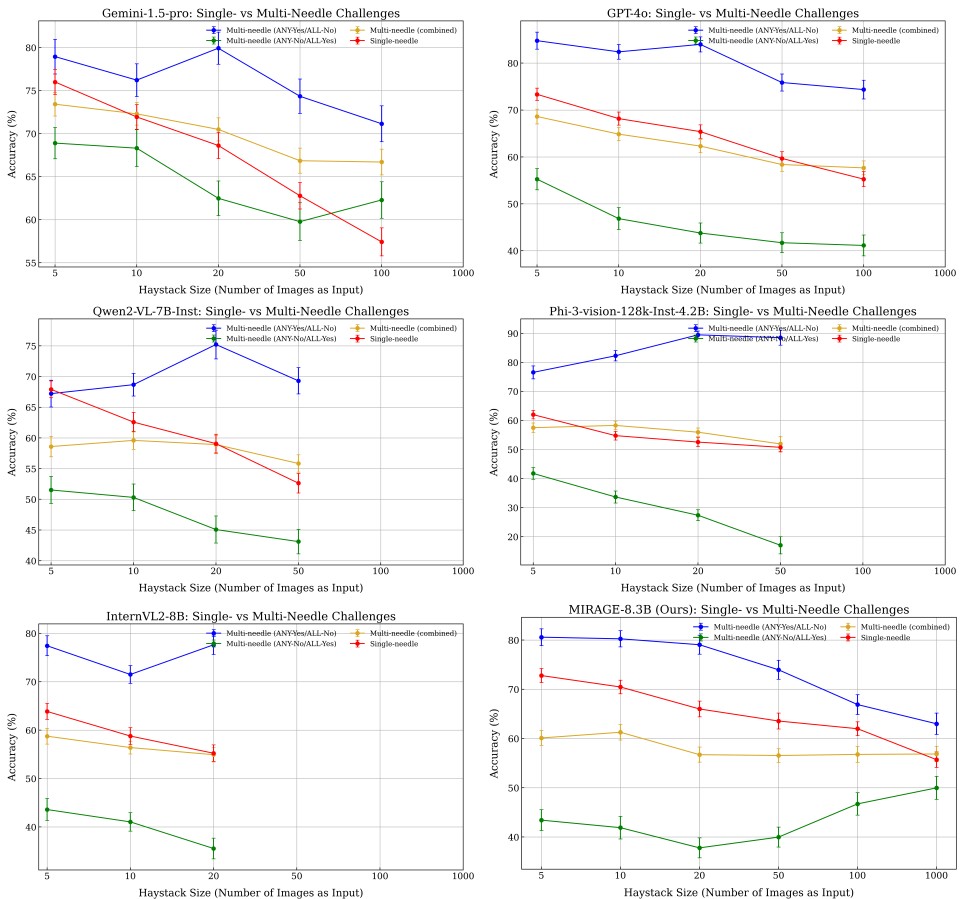

Figure C.3: We break down multi-needle challenges to two categories and compare that with the single-needle track. Please refer to subsection C.4 for detailed analyses and discussions.

## D  POTENTIAL LIMITATIONS AND SOCIETAL IMPACT

**VHs:** Just as the NIAH benchmark (easily solvable with regex) serves as a basic unit test for LLMs in long-context NLP tasks, VHs offers a similarly basic unit test for evaluating LMMs in long-context visual understanding, with the main focus on visual retrieval and reasoning across a large collection of images. While its simplicity and unprecedented scale are key strengths, the primary limitation of the benchmark is the scope: since it is based on MS-COCO images, it inherits the implicit biases in the dataset including notable gender, race, and location biases (Hendricks et al., 2018; Bhargava & Forsyth, 2019; Hirota et al., 2022; Wang et al., 2022). It is important to work in the future toward developing benchmarks that do not favor models that prefer data having such biases. Beyond such implicit biases, COCO objects are also limited to 80 categories - strong performance on the VHs benchmark does not imply that the model will generalize well to all MIQA problems. Finally, the VHs benchmark is primarily template-based, which means that it does not evaluate the language reasoning capabilities of the LLM. A more complex benchmark would require making more detailed inferences, and require multi-hop reasoning across a wider range of open-domain object sets.

**MIRAGE:** MIRAGE is significantly more efficient than its LLaVA base, while simultaneously performing better on many MIQA benchmarks. To perform well on MIQA benchmarks, however, we note that MIRAGE sacrifices some single-image performance, likely due to inefficiencies in the multi-task training setup. It remains interesting and necessary for future work to explore how such approaches can retain single-image performance while improving multi-image capabilities. It is also important to recognize that as a large multi-modal model, the potential for misuse of the model exists. Many of the impacts of such models are well studied in other related works (Bender et al., 2021; Liu et al., 2023b;a). Recognizing this, MIRAGE inherits the safety mechanisms from the LLaVA code-base (Liu et al., 2023b), and includes relevant training details in a fully public code release.

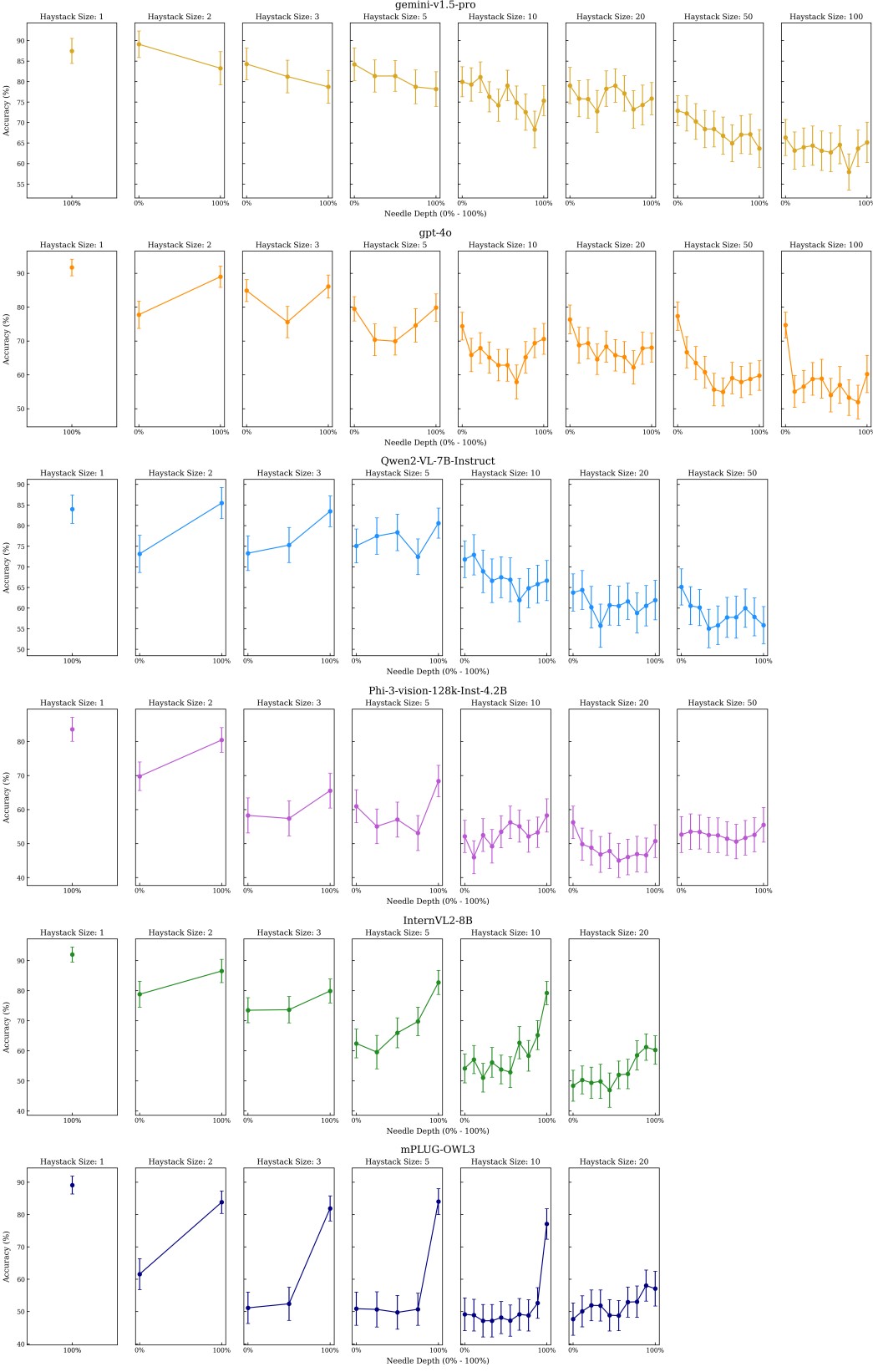

Figure C.4: Visualization of the positional bias pathology across various models.

