# OpenReview forum: "Visual Haystacks: A Vision-Centric Needle-In-A-Haystack Benchmark"
_ICLR.cc/2025/Conference — ICLR 2025 Poster_

### Official Review · Reviewer_io8N · 2024-11-01

**Soundness:** 3
**Presentation:** 2
**Contribution:** 2
**Rating:** 6
**Confidence:** 4

**Summary:**

This paper introduces a long context, visual needle in a haystack benchmark which composed of 1k yes/no questions changeling the model to reasoning and find the target object in the images. It evaluated on both open-source and closed-source LMMs and reveal several critical findings such as susceptibility to visual distractors, difficulty in multi-image reasoning, and a bias in image positioning. It introduces a new baseline called MIRAGE (Multi-Image Retrieval Augmented Generation) for better handling of VH tasks.

**Strengths:**

1. This paper introduced a new visual needle in a haystack benchmark which composed of 1k yes/no questions.
2. Evaluated on both open-source and close-source models and gained three insightful findings.
3. Introduced a new baseline called MIRAGE for better handling of visual haystack tasks.

**Weaknesses:**

1. The questions are only limited to yes/no questions.
2. The question template are very limited, seems only three.
3. MIRAGE has a significant performance drop in 4 out of 7 general VQA tasks.
4. The approach of MIRAGE, deselecting unrelated (distracting) images somehow circumvents the VH challenge, as the this challenge lies in how model can reasoning in long context.
5. The task of finding a target object seems still not simulating a real world scenario of long context visual reasoning task.

**Questions:**

1. I'm confused about the difference between the MIRAGE model in Table 1 and the Q-former Model in Table. Doesn't MIRAGE utilize Q-former.
2. See above.

---

> ### Author Response · Authors · 2024-11-18
> **Rebuttal by Authors**
>
> We appreciate the reviewer for providing thoughtful feedback and for highlighting our paper’s contributions, including the creation of a novel and insightful benchmark dataset and the development of the MIRAGE framework. Below, we respond to each of the concerns:
>
> **[W1, W2] Limited Question Types and Templates**
>
> Our VHs benchmark is intentionally designed with binary questions and simple templates. The rationale is to isolate and evaluate basic skills, visual retrieval and basic cross-image reasoning, without introducing confounding factors, like reasoning the logic in questions. By keeping the tasks straightforward, we provide a diagnostic unit test allowing researchers to pinpoint whether deficiencies lie in visual retrieval or reasoning. While more complex and diverse question templates are valuable for simulating real-world scenarios, they can obscure the diagnostic clarity of model performance. While expanding VHs to include richer templates is a logical next step, our results show that current LMMs struggle with even this simple test, so we strongly believe that such an expansion is out of scope for this initial effort. This point was briefly mentioned in L150-155 and Appendix D of our submission and further clarified in Appendix D of the updated paper.
>
> **[W3] Performance Drop in General VQA Tasks**
>
> We would like to emphasize that the primary focus of this paper is on large-scale multi-image QA tasks, where MIRAGE achieves SOTA results on the VHs benchmark (a unit test for these capabilities) and a real-world benchmark dataset, RetVQA, among all open-source solutions. While MIRAGE shows slightly reduced performance on some single-image QA tasks as detailed in Table 1, its main advantages are:
> 1. **Scalability**: MIRAGE is the only framework capable of processing thousands of images in a single query, greatly surpassing the input capacity of both proprietary and open-source models.
> 2. **Performance**: MIRAGE excels in multi-image tasks and, despite a slight drop in some single-image tasks like TextVQA, it outperforms LLaVA-v1.5-7B in some others like MMB(-CN), and rivals proprietary models in some cases.
> 3. **Efficiency**: MIRAGE achieves an 18x token compression and gets faster runtime in Table 6 (B). While this may slightly impact single-image QA performance, it is a strategic trade-off that enables MIRAGE to excel in large-scale multi-image tasks.
>
> This clarification was added to Section 5 of the updated paper.
>
> **[W4] MIRAGE’s Visual-RAG Approach**
>
> While MIRAGE functions as a visual-RAG framework with a retriever and an LMM, it is an end-to-end trained "single-model architecture," as detailed in Section 4.1 and Figure 5(A) of the submission paper. Thus, the "retriever" in this context should be understood as a component that primarily de-selects irrelevant details to enhance long-context reasoning. Based on this, we believe that our comparison with other “single-model LMMs” on VHs is both fair and valid. We have addressed this further in L310-318 of the updated paper.
>
> To assess the efficacy of the de-selection component, we compare the retrieval accuracy between MIRAGE and CLIP in Figure 6(A) and the QA accuracy between MIRAGE and CLIP+LLaVA in Table C.2 of the original submission (updated to Table C.3). These results highlight that such naive combinations fall short of optimal performance, emphasizing the significance of MIRAGE’s architecture.
>
> **[W5] Real-World Relevance of the VHs Benchmark**
>
> While the VHs benchmark serves as a diagnostic unit test (as mentioned in [W1]) and may not fully replicate real-world scenarios, we argue that the ability to find images containing specific objects is a foundational skill essential for various large-scale multi-image reasoning applications. These include searching for objects of interest in personal photo albums, identifying patterns in medical image databases, monitoring deforestation using satellite imagery, mapping urban changes with autonomous navigation data, and understanding consumer behavior through retail surveillance footage.
>
> This argument is addressed in the updated Lines 33–39 of the submission paper. We are happy to have further discussion if the reviewer could suggest specific tasks or datasets that represent real-world long-context challenges.
>
> **[Q1] Clarification on MIRAGE and Q-Former (Table 1 and 2)**
>
> To clarify, the implementation of Q-former is consistent across both tables and MIRAGE does utilize Q-former for image token compression. Table 1 reflects the full MIRAGE framework's performance, incorporating the Q-former compressor, a retriever module, and the Llama-v3.1 LLM. In contrast, Table 2 in the original submission (Table C.2 in the updated paper) presents an ablation study focused on Q-former's effectiveness in a simpler setup using LLaVA-v1.5 (lmsys/vicuna-v1.5-7b) without the retriever. Adjustments have been made to the paper to clarify these distinctions.

---

> > ### Comment · Reviewer_io8N · 2024-11-26
> >
> > Thank you for your response. The authors have addressed my major concerns. I'll maintain my score.

---

> > > ### Author Response · Authors · 2024-11-26
> > > **Official Comment by Authors**
> > >
> > > Thanks for the response. We will clarify all issues mentioned in the review/rebuttal in the final revision.

---

### Official Review · Reviewer_eUWk · 2024-11-02

**Soundness:** 3
**Presentation:** 3
**Contribution:** 2
**Rating:** 6
**Confidence:** 2

**Summary:**

This paper addresses the limitations of Large Multimodal Models (LMMs) in multi-image question answering, where handling large visual contexts does not ensure effective retrieval and reasoning across images. Current benchmarks reveal biases and challenges in MIQA, such as poor cross-image reasoning and sensitivity to information placement. To overcome these, the authors propose "Visual Haystacks (VHs)," a vision-centric benchmark that tests retrieval and reasoning over multiple images, highlighting models' struggles with visual distractors and multi-image reasoning. They also introduce MIRAGE, an open-source Multi-Image Retrieval Augmented Generation framework capable of handling up to 10,000 images on a single GPU, achieving significant improvements over existing models and setting new standards in MIQA benchmarks like RetVQA. Key contributions include VHs, systematic LMM evaluation, and MIRAGE's scalable MIQA capabilities.

**Strengths:**

- I generally feel the direction is important to our community where design meaningful Visual Haystack benchmark for evaluating VLM.
- Some interesting points are discovered when evaluating models on the proposed benchmark. Since random guess could achieve 50% accuracy in the proposed benchmark, some open-sourced VLMs performance significantly drop even the Haystack size is very small. However, those models maintain high scores in some public evaluation-datasets.
- Some detailed experiments are conducted such as needle position and running time.
- The proposed benchmark are made publicly available under MIT license, which is good for community.

**Weaknesses:**

- Benchmark construction is still mainly centered around recognition tasks, based on benchmark design principles listed in Line129~138. Basically, it requires a strong recognition among all the input images, rather than true visual reasoning.
- Based on the Figure 2 and 3, certain models, such as Gemini, GPT and the proposed MIRAGE, consistently perform better on the proposed multi-needle challenges compared to single-needle tasks. However, the multi-needle challenges are intentionally designed to be more difficult, as they demand additional reasoning across multiple images. Does this mean failure in designing the benchmark?
- Since the benchmark is constructed in way of examining recognition, therefore the proposed method contain ad-hoc modules, such as "a retriever module then calculates relevance scores, ensuring that only the most relevant images are passed to the LLM for final reasoning." Does this design hold for general visual reasoning tasks? For example, many of the tested single image dataset used in this paper, do not need this retriever module at all.
- The proposed framework achieved not-very-good performance on some of the tested datasets. Also, there are many datasets that not being tested such as SEED, MME, and CHAIR.

**Questions:**

- Could you please address the points raised in the above weakness?
- Could you please add some randomly sampled failure cases made by GPT or Gemini? Sometimes failure cases can tell more than good cases.
- Could you please address the ethics concerns around the code license?

**Details Of Ethics Concerns:**

Is it possible for the author to release their code under the MIT license, considering it is derived from the Apache 2.0-licensed LLaVA codebase? Could the author elaborate this point?

---

> ### Author Response · Authors · 2024-11-18
> **Rebuttal by Authors**
>
> We appreciate the reviewer for providing valuable comments and for highlighting our strengths in developing a meaningful visual NIAH benchmark, uncovering interesting findings when evaluating LMMs, and providing rigorous experiments. Below, we address each of the concerns:
>
> **[W1] Focus on Recognition Tasks**
>
> We want to clarify that object recognition—retrieving key visual content from a large collection of data—is a key skill for real-world long-context visual understanding tasks like searching through photo albums or analyzing medical and satellite imagery. Just as the NIAH benchmark (easily solvable with regex) serves as a basic unit test for LMMs, VHs—despite potentially being solvable with an object detector as shown in Figure 2 and 3 of the submission—can also act as a diagnostic tool for LMMs.
>
> Additionally, VHs assess cross-image reasoning, where LMMs often show significant performance declines when integrating information across multiple images, as detailed in Figure 3(A) and Appendix C.4. These points were briefly discussed in Appendix D of the submission and have been elaborated upon in the updated paper.
>
> **[W2] Multi-Needle Challenges and Dataset Design**
>
> We acknowledge that performance on the multi-needle track sometimes surpasses the single-needle one for image sets larger than 20. We'd like to clarify that the performance on the multi-needle track is not necessarily lower than that of the single-needle track. The multi-needle track poses questions in the format: "Q: For all images with <anchor object>, do ALL/ANY of them contain <target object>? A: Yes/No." We can then categorize each data point into two cases:
> 1. The ANY-Yes/ALL-No QA pairs: These cases are easier in the multi-needle track since retrieving at least one correct image can suffice. Conversely, the single-needle task demands precise retrieval, posing a greater challenge for models prone to false negatives.
> 2. The ANY-No/ALL-Yes QA pairs: These are more difficult in the multi-needle track than the single-needle one, as the model must retrieve all relevant images and integrate their information, demanding stronger cross-image reasoning.
>
> The results in Figure C.3 of the updated paper support the above explanation, showing that the benchmark effectively reveals nuanced model behaviors rather than indicating a design flaw. We have included this discussion in Appendix C.4.
>
> **[W3] Relevance of the Retriever Module**
>
> We believe that the retriever-LMM collaboration in our MIRAGE framework is well-motivated and is not ad-hoc in design.
> 1. Retriever: As there can be a lot of irrelevant visual content in the input, a retriever, filtering images based on relevance before reasoning occurs in the LMM, can make the whole system more efficient in terms of runtime (See Figure 6 (B) of the submission paper) and prevent the LMMs suffering from visual distractors – a limitation for current LMMs nowadays as shown in Figure 1 of the submission paper.
> 2. LMM: Given a small number of images, LMMs can be a fundamental and critical processor for the visual reasoning and next token prediction task for the actual answer generation.
>
> This framework makes MIRAGE robust for both the VHs unit tests and realistic large-scale multi-image QA tasks. These points were briefly addressed in Section 4.1 of the submission and further emphasized in L310-318 in the updated paper.
>
> **[W4] Performance on Other Benchmarks**
>
> The primary focus of the paper is on basic capabilities—visual retrieval and basic cross-image reasoning—for large-scale multi-image QA. MIRAGE achieves SOTA results on the VHs benchmark (a unit test for these capabilities) and on the existing RetVQA dataset among all open-source solutions, indicating its superiority in multi-image QA.
>
> In addition to the main focus, we've included seven common single-image QA benchmarks as a bonus. On them, MIRAGE performs on par with existing solutions as shown in Table 1. Updates for MME and SEED-Bench are also reflected in Table 1. For CHAIR [1], noted as a metric for hallucination detection, we have included an advanced hallucination benchmark POPE in Table 1 of the submission.
>
> [1] Rohrbach, Anna, et al. "Object hallucination in image captioning." EMNLP 2018.
>
> **[Q2] Analyses on Failure Cases**
>
> We have added an analysis of failure cases for Gemini v1.5 and GPT-4o in Figure C.2 of the updated paper. In summary, these models tend to struggle with small and non-salient objects due to the lack of a filtering module.
>
> **[Q3] Code Licensing Issues**
>
> Both the MIT and Apache 2.0 licenses are permissive and compatible with one another (i.e. neither is a copy-left license or restricts wide release). We plan to release our code under the MIT license and retain Apache 2.0 for any code originally licensed by LLaVA. The LLaVA LICENSE file was included in our anonymous repository to comply with Apache 2.0. We appreciate the reviewer's concern and are open to discussing any specific issues further.

---

> > ### Author Response · Authors · 2024-11-27
> > **Gentle Reminder**
> >
> > Dear Reviewer,
> >
> > As we are now midway through the discussion period, we wanted to send a friendly reminder about our response. We believe that our response, along with the revised version of the paper—particularly the clarification on VHs benchmark dataset design and MIRAGE's performance—addresses the key points raised in your review.
> >
> > If you have any further questions or concerns, we would be more than happy to address them during the discussion period. Thank you for your time and feedback.
> >
> > Best,
> >
> > The Authors

---

> > > ### Comment · Reviewer_eUWk · 2024-11-28
> > > **Reviewer response**
> > >
> > > I thank the efforts made by the author. The rebuttal address lots of my concerns.
> > >
> > > However, I remain my concerns around (1) the proposed benchmark requires a strong recognition among all the input images, rather than true visual reasoning; (2) the designed retriever module is ad-hoc and unnecessary to some tasks, where the author avoid to responding "many of the tested single image dataset used in this paper, do not need this retriever module at all.".
> > >
> > > I'd like to draw the attention to AC on this manner and open to discuss further with all other reviewers.
> > >
> > > Regarding the score, I prefer to increase my score and reduce my confidence score. Also, for a all-6-score paper, I recommend AC to pay attention to potential weak points. I will not fight for its acceptance.

---

> > > > ### Author Response · Authors · 2024-12-02
> > > > **Official Comment by Authors**
> > > >
> > > > Thank you for your thoughtful review and valuable feedback on our work, and your openness to further discussion. Regarding recognition vs. reasoning, VHs intentionally focuses on foundational skills as a diagnostic unit test to evaluate core model capabilities before tackling more complex tasks (though we agree, an iteration of the dataset could be expanded to far more complex visual reasoning tasks, perhaps in many different languages and cultural contexts).  While VHs may not fully simulate real-world scenarios, we believe its diagnostic nature offers important insights for advancing LMM capabilities. The retriever module in MIRAGE addresses efficiency and distractor challenges, a necessity for real-world large-scale tasks. While the retriever may be less useful in single image datasets, we believe that multimodal retrieval-augmented generation will be necessary in the future, and MIRAGE's retriever enables MIRAGE to be the first, and only, open-source LMM capable of handling more than 10,000 input images. Thank you once more for your time and thoughtful feedback; we deeply value your insights and look forward to refining our work further based on your suggestions.

---

### Official Review · Reviewer_kpNk · 2024-11-04

**Soundness:** 3
**Presentation:** 3
**Contribution:** 3
**Rating:** 6
**Confidence:** 4

**Summary:**

The authors presents Visual Haystacks (VHs), a new vision centric benchmark designed to assess the performance of Large Multimodal Models (LMMs) in the multi-image question answering (QA) task. In addition, the author proposes a new visaul-RAG framework, MIRAGE, to enhance the task performance.

**Strengths:**

* Novel Multi-Image QA Benchmark: The authors introduce an interesting multi-image QA benchmark, Visual Haystacks, designed around a vision-centric "needle-in-a-haystack" scenario, providing a fresh and challenging setting for the LMM evaluation.

* Comprehensive Model Evaluation:  The paper conducts a thorough evaluation of LMMs on the VHs benchmark, uncovering important insights into current models, such as vulnerability to visual distractors, challenges with multi-image understanding, and tendencies toward positional visual bias.

* Novel Visual RAG Framework: The authors introduce a novel visual RAG framework that combines a compressor and a retriever. The compressor efficiently processes up to 10,000 images on a single 40GB A100 GPU, while the retriever identifies the top-k most relevant images for a given question, enhancing the framework’s scalability and efficiency.

**Weaknesses:**

* Limited Object Diversity: The authors constructed the VHs benchmark using objects from the COCO dataset, which contains only 80 object categories. This limited selection may restrict the diversity and comprehensiveness of the benchmark, potentially affecting its ability to evaluate models across a broader range of visual scenarios.

* Restricted Question Diversity: The authors appear to rely on a few simple templates to generate questions, which may restrict the variety of question types in the benchmark.

* More like Object Detection than QA Reasoning: Many questions in the benchmark (e.g., "For the image with a truck, is there a dog?") seem to primarily assess the model’s object detection abilities rather than its visual QA reasoning skills. It is questionable if the benchmark requires the advanced visual QA reasoning skills from the models.

* Missing Related Work: The paper does not reference several recent multi-image QA benchmarks, for example:
 1. CompBench: A Comparative Reasoning Benchmark for Multimodal LLMs
 2. MANTIS: Interleaved Multi-Image Instruction Tuning
 3. MUIRBENCH: A Comprehensive Benchmark for Robust Multi-Image Understanding.

Additionally, a similar multi-image retrieval approach was introduced in "ColPali: Efficient Document Retrieval with Vision Language Models", but this work was also not cited.

**Questions:**

Please see the weakeness. In addition,
* How many templates were used to generate questions?
* What advantages does the VHs benchmark offer compared to recent multi-image QA benchmarks?

---

> ### Author Response · Authors · 2024-11-18
> **Rebuttal by Authors**
>
> We appreciate the reviewer for highlighting our paper’s contributions, including the construction of a novel benchmark dataset, the rigorous evaluation of various LMMs, and the development of an innovative visual-RAG framework. Below, we address each of the concerns:
>
> **[W1, W2, Q1] Limited Object and Question Diversity**
>
> While we understand that some of the questions are simple to human users, the primary objective of the Visual Haystacks (VHs) benchmark is to serve as a fundamental unit test for assessing LMMs' multi-image processing capabilities. As noted in L150-155 of both the submission paper and the updated paper, we deliberately constructed the dataset using COCO objects and straightforward (one for the single-needle track and two for the multi-needle track) templates to isolate and evaluate core retrieval and reasoning skills without additional confounding variables (such as out of domain data, as it is likely that most models are highly familiar with COCO objects).
>
> While we believe that expanding the object and question diversity is important (such as adding an open object set, adding questions that reason over actions and attributes, exploring multiple languages, or adding multi-step inference questions), our results presented in this paper demonstrate that models struggle with even this simple test, so we strongly believe that such an expansion is out of scope for this initial effort. We hope to consider such expansions for the following versions of the benchmark. These limitations and future direction are included in Appendix D of the paper, for both the updated version and the submission version.
>
> **[W3] More like Object Detection than QA Reasoning**
>
> We would like to clarify that detection—retrieving key information from a large collection of data—is a fundamental skill for enabling real-world long-context visual understanding tasks, like searching through photo albums or analyzing medical and satellite imagery in large databases. Just as the NIAH benchmark (easily solvable with regex) serves as a basic unit test for LLMs in long-context NLP tasks, we believe VHs offers a similarly essential unit test for evaluating LMMs in long-context visual understanding. While VHs could theoretically be addressed using an object detector (which we have already included as a baseline in Figure 2 and 3 of the submission), they represent a useful diagnostic tool/unit test for assessing these models as mentioned above. This point was briefly mentioned in L30-L50 and Appendix D of our submission and further clarified in Appendix D of the updated paper.
>
> Also, it's important to emphasize that our dataset is not solely focused on detection. It also includes a basic assessment of cross-image reasoning. In Figure 3 (A) of the submission paper and the updated Appendix C.4, we observe that LMMs experience significant performance degradation in the multi-needle track compared to the single-needle track, where models must integrate information across multiple images.
>
> As previously mentioned, VHs is designed to diagnose LMMs on two basic capabilities—visual retrieval and reasoning. We believe that expanding the scope of the benchmark, while valuable future work, is beyond the scope of this current contribution.
>
> **[W4] Missing Related Work**
>
> We thank the reviewer for pointing out these relevant contemporary works. We've adjusted the related work section to add these.
>
> **[Q2] The advantages of VHs over existing multi-image benchmarks**
>
> VHs offers two main advantages over existing multi-image benchmark datasets:
> 1. **Simplicity**: VHs is specifically designed to evaluate LMMs’ capabilities in visual retrieval and basic cross-image reasoning without introducing additional confounding factors, such as out-of-distribution images or complex language reasoning. By maintaining this simplicity, VHs serves as a diagnostic tool for LMMs, where the single-needle track focuses on visual retrieval performance, and the multi-needle track evaluates basic cross-image reasoning. In contrast, existing multi-image benchmarks often aim to address domain-specific or real-world applications, making the questions inherently more complex. While this complexity may reflect harder real-world challenges, it can make diagnosing specific LMM abilities difficult, as answering a single question often requires multiple intermixed capabilities.
> 2. **Scale**: Existing multi-image benchmarks like RETVQA include fewer than 30 images per question. The three datasets mentioned by the reviewer contain even fewer images, with fewer than 10 images per question. In comparison, VHs scales up to 10K images per question, far exceeding the size of existing benchmarks. This large-scale setting better mirrors real-world large-scale multi-image QA tasks, like photo album searching or analyzing medical and satellite imagery in large databases.
>
> We have clarified these points in the related work section of the updated paper. Thank you again for raising this issue.

---

> > ### Comment · Reviewer_kpNk · 2024-11-26
> >
> > Thank you for your response. The authors addressed my concerns, and I will maintain my score.

---

> > > ### Author Response · Authors · 2024-11-26
> > > **Official Comment by Authors**
> > >
> > > Thanks for the response. We will clarify all issues mentioned in the review/rebuttal in the final revision.

---

### Author Response · Authors · 2024-11-22
**Follow-Up on Rebuttal for "Visual Haystacks"**

Dear Reviewers,

We're following up on the rebuttal for our paper, "Visual Haystacks: A Vision-Centric Needle-In-A-Haystack Benchmark." We appreciate the time and effort you've invested in reviewing our work.

In our rebuttal and the updated paper, we've thoroughly addressed the concerns and suggestions raised in the initial reviews. If you find that all your questions have been resolved, we kindly ask you to consider reflecting this in the scores. Should you have any additional questions or require further clarification, we are eager to engage in more discussion during the official ICLR discussion period.

Thank you once again for your thoughtful feedback and consideration.

Best regards,

The Authors

---

### Meta-Review · Area_Chair_JzTu · 2024-12-23

**Metareview:**

The submission introduces a new benchmark "Visual Haystacks" for measuring the quality of large multimodal models (LMMs) at the task of multi-image reasoning. Along with the benchmark dataset, it also introduces a RAG framework (MIRAGE) that can process a magnitude larger number of images compared to prior work. This uses a retrieval-based relevance filter as well as compression of image features to fit into the context length of the LMMs.

After the initial round of reviews, this submission received scores of 6, 6, 6. The reviewers did not find sufficient reason to reject this submission during the discussion, and the consensus remained positive. The AC recommends acceptance, and requests the authors to use the constructive feedback from reviewers to update the submission.

**Additional Comments On Reviewer Discussion:**

During the rebuttal, reviewers raised concerns about the simplicity of the proposed benchmark - mostly consisting of binary answers and templated questions, as well as lack of diversity in concepts covered. The reviewers were satisfied with the explanation provided by the authors - that using simple questions helped remove confounding factors introduced by the need to understand complex questions, and focused purely on visual reasoning. Further, this benchmark focuses on common objects instead of exploring the long tail of visual concepts.
During the discussion, the reviewers did not feel that the shortcomings of the submission merited rejection.

---

### Decision · Program_Chairs · 2025-01-22

Accept (Poster)